# The Prognostic Value of Neutrophil-to-Lymphocyte Ratio and Platelet-to-Lymphocyte Ratio in Patients with Hepatocellular Carcinoma Receiving Atezolizumab Plus Bevacizumab

**DOI:** 10.3390/cancers14020343

**Published:** 2022-01-11

**Authors:** Jing-Houng Wang, Yen-Yang Chen, Kwong-Ming Kee, Chih-Chi Wang, Ming-Chao Tsai, Yuan-Hung Kuo, Chao-Hung Hung, Wei-Feng Li, Hsiang-Lan Lai, Yen-Hao Chen

**Affiliations:** 1Division of Hepatogastroenterology, Department of Internal Medicine, Kaohsiung Chang Gung Memorial Hospital and Chang Gung University College of Medicine, Kaohsiung 833, Taiwan; wajing@cgmh.org.tw (J.-H.W.); keekkm@cgmh.org.tw (K.-M.K.); tony0779@cgmh.org.tw (M.-C.T.); 0104kuo@cgmh.org.tw (Y.-H.K.); 4366hung@cgmh.org.tw (C.-H.H.); 2Division of Hematology-Oncology, Department of Internal Medicine, Kaohsiung Chang Gung Memorial Hospital and Chang Gung University College of Medicine, Kaohsiung 833, Taiwan; chenyy@cgmh.org.tw (Y.-Y.C.); bbaroma@cgmh.org.tw (H.-L.L.); 3Division of General Surgery, Department of Surgery, Kaohsiung Chang Gung Memorial Hospital and Chang Gung University College of Medicine, Kaohsiung 833, Taiwan; chihchiwang@cgmh.org.tw (C.-C.W.); webphone@cgmh.org.tw (W.-F.L.); 4School of Medicine, College of Medicine, Chang Gung University, Taoyuan 333, Taiwan; 5Department of Nursing, Meiho University, Pingtung 912, Taiwan; 6School of Medicine, Chung Shan Medical University, Taichung 402, Taiwan

**Keywords:** atezolizumab, bevacizumab, hepatocellular carcinoma, neutrophil-to-lymphocyte ratio, platelet-to-lymphocyte ratio

## Abstract

**Simple Summary:**

Atezolizumab plus bevacizumab has been approved as the first-line systemic treatment for unresectable hepatocellular carcinoma (uHCC) patients. However, the real-world practice of this combination is limited. We reported 48 uHCC patients who received atezolizumab plus bevacizumab, the median progression-free survival (PFS) was 5.0 months, and the objective response rate and disease control rate were 27.1% and 68.8%, respectively. The severity of most adverse events was predominantly grade 1–2, and most patients tolerated the toxicities. We also used inflammatory biomarkers to predict PFS, including neutrophil-to-lymphocyte ratio (NLR) and platelet-to-lymphocyte ratio (PLR). Univariate and multivariate analyses revealed NLR and PLR were independent prognostic factors for superior PFS. The significance of our study is the first research to investigate the prognostic value of NLR and PLR among uHCC patients receiving atezolizumab plus bevacizumab. It would bring more information to physicians about the efficacy and safety of atezolizumab plus bevacizumab in real-world clinical practice.

**Abstract:**

Atezolizumab plus bevacizumab has been approved as the first-line systemic treatment for patients with unresectable hepatocellular carcinoma (uHCC). This study was designed to assess the clinical impact of atezolizumab plus bevacizumab in uHCC patients. A total of 48 uHCC patients receiving atezolizumab plus bevacizumab were identified, including first-line, second-line, third-line, and later-line settings. In these patients, the median progression-free survival (PFS) was 5.0 months, including 5.0 months for the first-line treatment, not reached for the second-line treatment, and 2.5 months for the third line and later line treatment. The objective response rate and disease control rate to atezolizumab plus bevacizumab were 27.1% and 68.8%, respectively. The severity of most adverse events was predominantly grade 1–2, and most patients tolerated the toxicities. The ratios of the neutrophil-to-lymphocyte ratio (NLR) and platelet-to-lymphocyte (PLR) were used to predict PFS in these patients. The optimal cutoff values of NLR and PLR were 3 and 230, and NLR and PLR were independent prognostic factors for superior PFS in the univariate and multivariate analyses. Our study confirms the efficacy and safety of atezolizumab plus bevacizumab in uHCC patients in clinical practice and demonstrates the prognostic role of NLR and PLR for PFS in these patients.

## 1. Introduction

Hepatocellular carcinoma (HCC) is globally one of the most common cancers and the second leading cause of cancer-related deaths in Taiwan [1]. Sorafenib and lenvatinib have been approved as first-line systemic treatment in patients with unresectable HCC (uHCC) who are not feasible for surgical intervention or other locoregional therapies, such as trans-arterial chemoembolization (TACE) or radiofrequency ablation (RFA) [2,3,4]. The SHARP and Asia-Pacific trials have confirmed the overall survival (OS) benefit of sorafenib in these uHCC patients compared to the placebo group [2,4]. According to the REFLECT trial, lenvatinib is non-inferior to sorafenib with respect to OS, and is better than sorafenib with respect to progression-free survival (PFS) and objective response rate (ORR) [3]. Recently, the IMbrave150 trial, a global, open-label, phase 3 study has shown that atezolizumab plus bevacizumab prolongs PFS and OS than does sorafenib in patients with uHCC [5]. In addition, the ORR of atezolizumab plus bevacizumab was approximately 30%, and the percentage of adverse events was comparable to that of sorafenib. Therefore, atezolizumab plus bevacizumab is becoming the preferred first-line systemic treatment against uHCC in clinical practice. 

Increasing evidence has revealed that chronic inflammation plays a predominant role in the process of tumor progression, including cancer cell proliferation, angiogenesis, and metastasis [6,7]. Tumor-associated neutrophils (TANs) have two different phenotypes: N1 (anti-tumorigenic) and N2 (protumorigenic), and high infiltration with N2 TANs has contributed to tumor cell proliferation, distant metastasis and poor prognosis [8]. The neutrophil-to-lymphocyte ratio (NLR) and platelet-to-lymphocyte ratio (PLR) are easy-to-obtain and cost-effective biomarkers that are widely used to predict treatment response and prognosis in several cancer types [9,10,11,12,13,14,15,16,17]. In addition, a review article summarizes the current evidence on the role of TANs in the pathogenesis and progression of HCC, and highlights the significance of NLR as a reliable biomarker with prognostic potential for HCC [8]. Several studies have confirmed the role of NLR and PLR in HCC [18]. NLR and PLR have been reported to be useful prognostic factors for predicting outcomesin patients with HCC who underwent hepatectomy [19,20]. On the other hand, NLR and PLR are regarded as independent markers of poor prognosis in HCC patients who received TACE or RFA [21,22,23]. In HCC patients with liver transplantation, elevated NLR and PLR are associated with early tumor recurrence [24,25]. Liu reported a meta-analysis that demonstrated the role of NLR and PLR in patients with HCC who were receiving sorafenib; patients with a lower baseline NLR and PLR had better response to sorafenib and superior OS compared to those with a higher NLR [26]. Another Japanese study revealed that low NLR was independently associated with better PFS, OS, and disease control in patients with HCC who received lenvatinib [27]; PLR could be used to predict OS in patients with uHCC who received lenvatinib [28]. Recently, NLR and PLR were reported to have strong predictive roles in patients with HCC who were treated with anti–PD-1 therapy, such as nivolumab [29].

Several studies have confirmed the efficacy and safety of sorafenib and lenvatinib in patients with HCC [30,31,32,33,34,35]. Nevertheless, information on atezolizumab plus bevacizumab in real-world practice is relatively limited. Recently, Iwamoto reported the first real-world outcomes of atezolizumab plus bevacizumab treatment in patients with uHCC in Japan [36]. However, to the best of our knowledge, information regarding the clinical impact of NLR in patients with uHCC receiving atezolizumab plus bevacizumab is unclear. The present study was designed to explore the efficacy and safety of atezolizumab plus bevacizumab and the prognostic significance of the NLR and PLR in patients with uHCC.

## 2. Materials and Methods

### 2.1. Study Population

We retrospectively reviewed patients with uHCC who received atezolizumab plus bevacizumab at Kaohsiung Chang Gung Memorial Hospital between January 2020 and October 2021. The eligibility criteria were as follows: (1) no evidence of a second malignancy or concurrent cholangiocarcinoma; (2) treatment with atezolizumab plus bevacizumab for more than 2 cycles; (3) follow-up duration > 4 weeks; (4) no esophageal or gastric varices detected by upper gastrointestinal endoscopy; and (5) precise collection of clinical data. Finally, 48 patients who were treated with atezolizumab plus bevacizumab were included in the study. 

The NLR was calculated by dividing the absolute neutrophil count by the absolute lymphocyte count measured in peripheral blood, and PLR was calculated as the ratio of absolute platelet count to absolute lymphocyte count. The receiver operating characteristic (ROC) curve analysis was used to identify the optimal cut-off values of NLR and PLR according to the Youden index (Youden Index = Sensitivity + Specificity − 1, range from 0 to 1), which is a commonly used measure of overall diagnostic effectiveness [37]. A cutoff value of 3 used for NLR, and a cutoff of 230 used for the PLR group were both reported by previous literature [29,38,39]. In addition, albumin-bilirubin (ALBI) score was calculated based on serum albumin and total bilirubin values using the following formula: ALBI score = (log_10_ bilirubin [μmol/L] × 0.66) + (albumin [g/L] × −0.085). The ALBI score was categorized into grade 1 (−2.60 or less), grade 2 (−2.59 to −1.39), or grade 3 (greater than −1.39).

### 2.2. Treatment Protocol and Safety Assessment

In our study, patients received atezolizumab at a dose of 1200 mg and bevacizumab at a dose of 5–7.5 mg/kg intravenously every 3 weeks. Treatment was continued until disease progression or the development of intolerable AEs. Patients were followed up at the outpatient clinic for assessment of AEs every 3 weeks, and the grade of AEs was assigned based on the National Cancer Institute Common Terminology Criteria for Adverse Events (CTCAE) version 5.0 [40]. 

### 2.3. Staging and Evaluation of Response

HCC diagnosis was based on pathological findings or according to the non-invasive criteria of the American Association for the Study of Liver Disease (AASLD) guidelines [41,42]. HCC was staged using the Barcelona Clinic Liver Cancer (BCLC) staging classification at the time of atezolizumab plus bevacizumab initiation [43]. Each patient must have had at least one measurable target lesion to evaluate treatment response using dynamic computed tomography (CT) or magnetic resonance imaging (MRI) of the liver every 9 weeks after commencement of treatment. The response was independently determined by two radiologists without any clinical information, in accordance with the guidelines of the Response Evaluation Criteria in Solid Tumors (RECIST) version 1.1 [44]. 

### 2.4. Statistical Analysis

All data analyses were performed using the SPSS 19 software (IBM, Armonk, NY, USA). The chi-square test was used for categorical variables. PFS was determined from the date of atezolizumab plus bevacizumab initiation to disease progression or death due to any cause. Actuarial analysis of cumulative survival was performed using the Kaplan–Meier method, and the differences were assessed with the log-rank test. Hazard ratios (HRs) with 95% confidence intervals (CIs) and *p*-values were calculated to quantify the strength of the associations between the prognostic parameters and survival. Parameters significantly associated with PFS in the univariate analysis were selected as covariates for multivariate Cox proportional hazards models. Statistical significance was set at *p* < 0.05. 

### 2.5. Ethics Statement

The present study was conducted in accordance with the Declaration of Helsinki. The Institutional Review Board of Chang Gung Medical Foundation approved this study (202101199B0) and waived the requirement for written informed consent owing to the retrospective design of this study. 

## 3. Results

### 3.1. Characteristics of Patients

Our study cohort consists of 48 patients with uHCC who received atezolizumab plus bevacizumab at Kaohsiung Chang Gung Memorial Hospital between January 2020 and October 2021, including 38 men and 10 women with a median age of 62 years (range: 31–80 years). The characteristics of these patients were documented at the time of atezolizumab plus bevacizumab administration. All patients had an Eastern Cooperative Oncology Group Performance Status (ECOG PS) score of 0 or 1, and all were classified as BCLC staging classification C. Most of these patients had Child–Pugh classification A (87.5%), and only 6 (12.5%) patients had Child–Pugh classification B. In terms of viral hepatitis, hepatitis B virus (HBV) infection was reported in 28 (58.3%) patients and hepatitis C virus (HCV) infection in 13 (27.1%) patients. The percentages of albumin-bilirubin (ALBI) 1 and 2 were similar (47.9% vs. 52.1%). At the time of analysis, the median follow-up period was 9.5 months for all 48 patients (range: 2.4–23.8). The demographic characteristics of the patients are presented in Table 1.

### 3.2. Assessment of the Cut-Off Value of NLR and PLR 

The optimal cutoff values of NLR and PLR were determined by ROC analysis; the ideal cutoff values for NLR and PLR were 3 and 230, respectively. According to the ROC curves, the area under the curve (AUC) for NLR and PLR was 0.782 (95% CI: 0.651–0.914, *p* = 0.001) and 0.777 (95% CI: 0.630–0.924, *p* = 0.001), respectively (Figure 1).

### 3.3. Efficacy Analyses of Atezolizumab plus Bevacizumab

The response to atezolizumab plus bevacizumab treatment was determined based on the RECIST criteria version 1.1, including 13 (27.1%) patients with partial response (PR), 20 (41.7%) with stable disease (SD), and 15 (31.2%) with progressive disease (PD), indicating a disease control rate (DCR) of 68.8%. The PFS was 9.6 months, 7.6 months, and 2.4 months in patients with PR, SD, and PD, respectively (*p* < 0.001). 

In addition, atezolizumab/bevacizumab was used as first-line treatment in 27 (56.2%) patients, second-line treatment in 12 (25.0%) patients, and third-line and later line treatment in 9 (18.8%) patients. The median PFS was 5.0 months for the first-line treatment, NR for the second-line treatment, and 2.5 months for the third line and later line treatment groups (*p* = 0.042).

In the first-line setting, the ORR and DCR were 29.6% and 66.6%, respectively; there was an ORR of 25.0% and DCR of 83.3% in the second-line group; even in the third line and later lines, the PR was 22.2% and DCR was 55.5%. The results of the survival analyses and treatment effects of atezolizumab plus bevacizumab are presented in Table 2.

In our study, the median PFS of the whole population was 5.0 months (Figure 2). In the analysis of PFS, there were no significant differences in all parameters in the univariate analysis, except for ECOG PS, AFP > 400 ng/mL, NLR, and PLR. The 31 patients who had ECOG PS 0 had better PFS than those with ECOG PS 1 (9.6 months versus 2.6 months, *p* = 0.004); superior PFS was noted in 27 patients with AFP < 400 ng/mL compared to the remaining 21 patients with AFP ≥ 400 ng/mL (9.6 months versus 2.8 months, *p* = 0.002). The patients with NLR < 3 were found to have longer PFS in compassion with those with NLR ≥ 3 (9.6 months versus 2.9 months, *p* = 0.009, Figure 3A); patients who had a PLR < 230 had a longer PFS than those who had PLR ≥ 230 (9.3 months versus 2.4 months, *p* = 0.001, Figure 3B). Multivariate analysis showed that no hepatectomy before atezolizumab plus bevacizumab (*p* = 0.019; HR, 0.39; 95% CI, 0.18–0.86), AFP < 400 ng/mL (*p* = 0.001; HR, 0.24; 95% CI, 0.11–0.54), NLR < 3 (*p* = 0.019; HR, 0.34; 95% CI, 0.14–0.84), and PLR < 230 (*p* = 0.014; HR, 0.36; 95% CI, 0.16–0.81) were independent prognostic factors for superior PFS. The univariate and multivariate analyses of PFS were shown in Table 3.

### 3.4. Safety Analyses

Most patients (93.8%) experienced AEs following atezolizumab plus bevacizumab treatment. The most common AEs were aspartate/alanine aminotransferase increase (85.4%), followed by proteinuria (35.4%), fatigue (25.0%), hypertension (22.9%), decreased appetite (22.9%), abdominal pain (18.8%), and nausea (14.6%). The severity of most AEs was grade 1–2; grade 3–4 toxicities were relatively rare, including aspartate/alanine aminotransferase increase (20.8%), hypertension (6.3%), proteinuria (4.2%), and diarrhea (2.0%). There were no drug-related grade 5 AEs. Most patients tolerated the AEs of atezolizumab plus bevacizumab, and none of the patients experienced treatment interruption or dose adjustment due to AEs. The frequencies of drug-related AEs are listed in Table 4. 

### 3.5. Case Presentation

Case 1: The 53-year-old man had past history of HBV and liver cirrhosis, and has been diagnosed with HCC in February 2021. MRI of liver revealed multiple liver tumors over S5/S7/S8 with left, right and main portal vein thrombosis. Then he received atezolizumab plus bevacizumab since February 2021; image was followed after completion of three cycles of this combination therapy. MRI of liver demonstrated that decreased in size of liver tumors and portal vein thrombosis, indicating PR (Appendix A).

Case 2: The 61-year-old woman has been HBV related liver cirrhosis and splenomegaly, and HCC was diagnosed in January 2021. MRI of liver showed liver tumors over S4/S7/S8 and enlarged lymph nodes over gastrohepatic and precaval regions. In addition, multiple bone metastasis was detected by bone scan. After that, she received atezolizumab plus bevacizumab for 5 cycles, and stable condition of liver tumors was mentioned by followed MRI of liver. Bone scan also revealed mild regressive change of bone metastasis. The treatment response was regarded as SD (Appendix A).

Case 3: The 31-year-old man has been diagnosed with HCC with lung metastasis in September 2020. CT of liver showed liver tumor over S7 and CT of chest revealed one small nodule over right upper lobe. Atezolizumab plus bevacizumab were given since September 2020 with a total of three cycles. Followed CT of liver demonstrated increased in size of liver tumor over S7, accompanied with multiple new liver nodules over both lobes, suggesting progression. On the other hand, markedly progressive change of metastatic lung nodule over right upper lobe was also mentioned on CT of chest. In conclusion, the response to atezolizumab plus bevacizumab was PD (Appendix A).

## 4. Discussion

Our study showed real-world evidence of the efficacy and safety of atezolizumab plus bevacizumab in patients with uHCC. Patients who used this combination therapy as first-line, second-line, or third-line therapy were enrolled. The ORR and DCR were 27.1% and 68.8%, respectively. The median PFS was 5.0 months for all patients, including 5.0 months for first-line use, NR for second-line use, and 2.5 months for third-line and later line use. In addition, we also reported the clinical predictors of NLR and PLR in these patients; higher NLR and PLR were independent prognostic factors of worse PFS in patients with uHCC who received atezolizumab plus bevacizumab. Most AEs of atezolizumab plus bevacizumab were grade 1–2, and most patients tolerated the toxicities. In conclusion, we showed the clinical efficacy and safety of atezolizumab plus bevacizumab and reveal the prognostic value of NLR and PLR for patients with uHCC in real-world clinical practice.

In our study, the dose of bevacizumab was different from that in the IMbrave150 trial. In the IMbrave150 trial, bevacizumab was administered at a dose of 15 mg/kg every 3 weeks; however, we only prescribed bevacizumab at a dose of 5 or 7.5 mg/Kg in our study. Bevacizumab is an anti-angiogenic agent with additional immunomodulatory effects, including normalizing tumor vasculature, increasing T-cell infiltration, decreasing the activity of immunosuppressive cells, and promoting the maturation of dendritic cells [45,46]. Bevacizumab may enhance the efficacy of atezolizumab by reversing vascular endothelial growth factor (VEGF)-mediated immunosuppression. Therefore, bevacizumab dose may not be fixed. Bevacizumab was prescribed with irinotecan/5-fluouracil at a dose of 5 mg/kg or 10 mg/kg every 2 weeks for patients with colorectal cancer (CRC) in a randomized phase III EAGLE study [47]. The median PFS was similar for both doses of bevacizumab, and no clear clinical benefit was noted in the high-dose bevacizumab group. In addition, Gordon et al. reported that free serum concentrations of VEGF could drop below detectable limits even when the dose of bevacizumab was as low as 0.3 mg/kg, indicating that a higher dose of bevacizumab may not be necessary for optimal activity in cancer treatment [48]. 

NLR and PLR have been proven to be associated with disease progression, tumor recurrence, and clinical outcome in several cancer types. Recently, a review article summarizes the current evidence on the significance of NLR in the pathogenesis and progression of HCC, and highlights the role of NLR as a reliable biomarker in the potential involvement in tumor therapy for HCC, such as immunotherapy, chemotherapy or liver transplantation [8]. Sorafenib and lenvatinib have been approved for first-line systemic treatment in patients with uHCC, and several studies have focused on the clinical utility of NLR and PLR in tracking treatment response in patients with HCC who received sorafenib or lenvatinib [26,27,28]. However, the prognostic value of NLR and PLR in patients receiving atezolizumab plus bevacizumab remained unclear. Our study is the first to report an association between NLR/PLR and clinical outcomes in this group. Patients with NLR < 3 had better PFS than those with NLR ≥ 3 (9.6 months versus 2.9 months); superior PFS was also mentioned in patients with PLR < 230 than in those with PLR ≥ 230 (9.3 months versus 2.4 months). Therefore, the results of the current study establish the clinical utility of NLR and PLR as biomarkers for tracking PFS in patients with uHCC receiving atezolizumab plus bevacizumab.

Recently, Iwamoto et al. reported the first real-world outcomes of atezolizumab plus bevacizumab treatment in 60 patients with HCC in Japan [36]. In our study, there were 27 patients who received atezolizumab plus bevacizumab as first-line treatment with a PR of 29.6% and DCR of 66.6%; these data were similar to the results of the IMbrave150 study. However, information about this combination therapy in the second-line or later line setting was unclear. Our study enrolled 20 and 15 patients with uHCC who received atezolizumab plus bevacizumab as second-line, or third-line and later lines, respectively. In the second-line setting, the ORR and DCR were 25.0% and 83.3%, respectively; in the third line and later lines, ORR and DCR were 22.2% and 55.5%, respectively. The median PFS was NR in the second-line group and 2.5 months in the third-line and later line groups, respectively. The clinical outcomes of atezolizumab plus bevacizumab in the second-line and later line settings were acceptable. To the best of our knowledge, this is the first study to present the clinical efficacy of atezolizumab plus bevacizumab in second-line and later line settings. 

Our study enrolled 20 and 15 patients with uHCC who received atezolizumab plus bevacizumab as second-line, or third-line and later lines, respectively. In the second-line setting, the ORR and DCR were 25.0% and 83.3%, respectively; in the third line and later lines ORR and DCR were 22.2% and 55.5%, respectively. The median PFS was NR in the second-line group and 2.5 months in the third-line and later line groups, respectively. The clinical outcomes of atezolizumab plus bevacizumab in the second-line and later line settings were acceptable. To the best of our knowledge, this is the first study to present the clinical efficacy of atezolizumab plus bevacizumab in second-line and later line settings. 

Our study has some limitations. First, the sample size of patients with receiving atezolizumab plus bevacizumab was relatively small; therefore, some prognostic factors of PFS may be difficult to identify. Second, the duration of the follow-up period may not have been long enough, therefore OS could not be evaluated. Third, there were different treatment strategies for atezolizumab plus bevacizumab, including first-line, second-line, third-line, and later lines, resulting in the relatively small population in the different groups. Therefore, there may be biases in the results of PFS. However, to the best of our knowledge, this is one of the limited studies designed to assess the clinical efficacy and safety, and the first cohort study designed to evaluate the prognostic role of NLR and PLR in patients who received atezolizumab plus bevacizumab in real-world practice. 

## 5. Conclusions

Our study confirms the efficacy and safety of atezolizumab plus bevacizumab in patients with uHCC in clinical practice and identifies a prognostic value of NLR and PLR for predicting PFS in these patients.

## Figures and Tables

**Figure 1 cancers-14-00343-f001:**
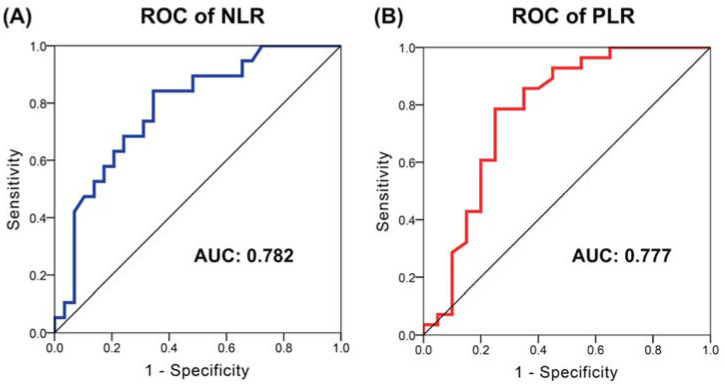
The receiver operating characteristic (ROC) curves of the NLR and PLR in patients with unresectable HCC who received atezolizumab plus bevacizumab. The area under the curves (AUC) for NLR (**A**) and for PLR (**B**).

**Figure 2 cancers-14-00343-f002:**
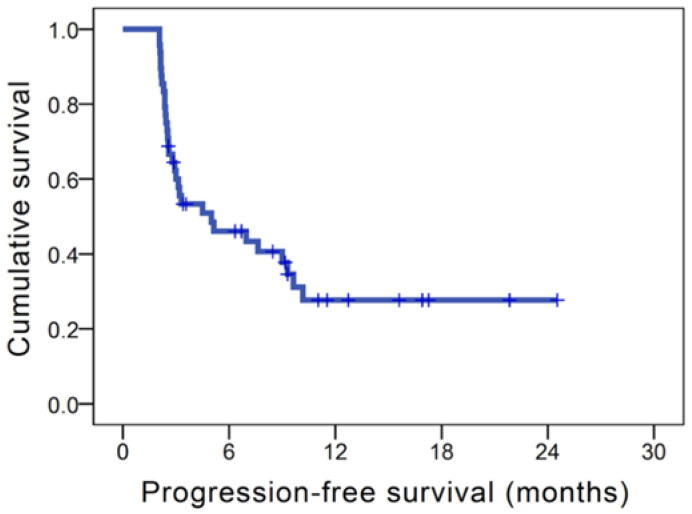
Kaplan–Meier survival curves of progression-free survival (PFS) in patients with unresectable hepatocellular carcinoma who received atezolizumab plus bevacizumab.

**Figure 3 cancers-14-00343-f003:**
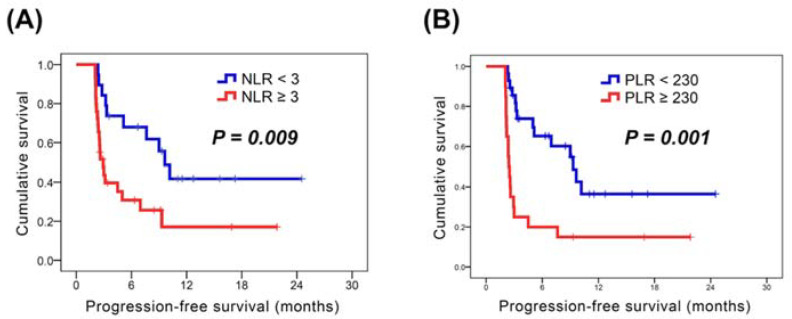
Kaplan–Meier survival analyses (**A**) Kaplan–Meier survival curves for patients with NLR ≥ 3 vs. those with NLR < 3; (**B**) Kaplan–Meier survival curves for patients with PLR ≥ 230 vs. those with PLR < 230.

**Table 1 cancers-14-00343-t001:** Characteristics of 48 patients with unresectable hepatocellular carcinoma who received atezolizumab plus bevacizumab.

Variable	Patient Number (%)
Age (median, range)	62 years old (31–80)
Sex	
Male	38 (79.2%)
Female	10 (20.8%)
ECOG PS	
0	31 (64.6%)
1	17 (35.4%)
Child–Pugh classification	
A	42 (87.5%)
B	6 (12.5%)
BCLC classification	
C	48 (100.0%)
ALBI grade	
1	23 (47.9%)
2	25 (52.1%)
Viral hepatitis status	
Hepatitis B	28 (58.3%)
Hepatitis C	13 (27.1%)
No	7 (14.6%)
Macrovascular invasion	
Yes	26 (54.2%)
No	22 (45.8%)
Main portal vein thrombosis	
Yes	9 (18.8%)
No	39 (81.2%)
Hepatectomy before atezolizumab plus bevacizumab	
Yes	18 (37.5%)
No	30 (62.5%)
Lymph node metastasis at the time of atezolizumab plus bevacizumab	
Yes	12 (25.0%)
No	36 (75.0%)
Extrahepatic spread at the time of atezolizumab plus bevacizumab	
Yes	26 (54.2%)
No	22 (45.8%)
AFP at the time of atezolizumab plus bevacizumab (median, range) ng/mL	157.1 (2.1->80000)

ECOG PS: Eastern Cooperative Oncology Group Performance Status; BCLC: Barcelona-Clinic Liver Cancer; ALBI: Albumin-Bilirubin; AFP: alpha-fetoprotein.

**Table 2 cancers-14-00343-t002:** Analysis of progression-free survival (PFS) according to the response rates and treatment lines of atezolizumab plus bevacizumab.

Variables	Number of Patients	PFS (Months)	*p* Value
Treatment response			
Partial response	13 (27.1%)	9.6	<0.001 *
Stable disease	20 (41.7%)	7.6	
Progressive disease	15 (31.2%)	2.4	
Treatment lines			
First line	27 (56.2%)	5.0	0.042 *
Second line	12 (25.0%)	NR	
Third line and later lines	9 (18.8%)	2.5	
Treatment lines	Partial response	Stable disease	Disease control rate
First line (N = 27)	8 (29.6%)	10 (37.0%)	18 (66.6%)
Second line (N = 12)	3 (25.0%)	7 (58.3%)	10 (83.3%)
Third and later lines (N = 9)	2 (22.2%)	3 (33.3%)	5 (55.5%)

PFS: progression-free survival; NR: not reached. * Statistically significant.

**Table 3 cancers-14-00343-t003:** Univariate and multivariate analyses of progression-free survival (PFS) in 48 patients with unresectable hepatocellular carcinoma who received atezolizumab plus bevacizumab.

Characteristics	No. of Patients	Univariate		Multivariate	
		PFS (Months)	*p* Value	HR (95% CI)	*p* Value
Age					
<60 years	21 (43.8%)	5.0	0.45		
≥60 years	27 (56.2%)	4.5			
Sex					
Male	38 (79.2%)	5.1	0.99		
Female	10 (20.8%)	3.0			
ECOG PS					
0	31 (64.6%)	9.6	0.004 *		
1	17 (35.4%)	2.6			
Child–Pugh classification					
A	42 (87.5%)	5.0	0.63		
B	6 (12.5%)	4.5			
Treatment lines					
First line	27 (56.2%)	5.0	0.60		
Second and later lines	21 (43.8%)	3.2			
ALBI grade					
1	23 (47.9%)	5.0	0.40		
2	25 (52.1%)	4.5			
Hepatitis B					
Yes	28 (58.3%)	5.0	0.79		
No	20 (41.7%)	3.3			
Hepatitis C					
Yes	13 (27.1%)	5.1	0.46		
No	35 (72.9%)	5.0			
Macrovascular invasion					
Yes	26 (54.2%)	3.3	0.23		
No	22 (45.8%)	7.6			
Main portal vein thrombosis					
Yes	9 (18.8%)	5.1	0.24		
No	39 (81.2%)	3.2			
Hepatectomy before atezolizumab plus bevacizumab					
Yes	18 (37.5%)	2.8	0.06		
No	30 (62.5%)	7.6		0.39 (0.18–0.86)	0.019 *
Lymph node metastasis at the time of atezolizumab plus bevacizumab					
Yes	12 (25.0%)	3.1	0.30		
No	36 (75.0%)	5.1			
Extrahepatic spread at the time of atezolizumab plus bevacizumab					
Yes	26 (54.2%)	3.3	0.92		
No	22 (45.8%)	5.1			
AFP ≥ 400 at the time of atezolizumab plus bevacizumab					
Yes	21 (43.8%)	2.8	0.002 *		
No	27 (56.2%)	9.6		0.24 (0.11–0.54)	0.001 *
NLR					
≥3	29 (60.4%)	2.9	0.009 *		
<3	19 (39.6%)	9.6		0.34 (0.14–0.84)	0.019 *
PLR					
≥230	20 (41.7%)	2.4	0.001 *		
<230	28 (58.3%)	9.3		0.36 (0.16–0.81)	0.014 *

ECOG PS: Eastern Cooperative Oncology Group Performance Status; HR: hazard ratio; CI: confidence interval; ALBI: Albumin-Bilirubin; AFP: alpha-fetoprotein; NLR: neutrophil-to-lymphocyte ratio; PLR: platelet-to-lymphocyte ratio. * Statistically significant, *p* < 0.05.

**Table 4 cancers-14-00343-t004:** The treatment-related adverse events in the 48 patients with unresectable hepatocellular carcinoma who received atezolizumab plus bevacizumab.

Adverse Event	Any Grades	Grade 3/4
Hypertension	11 (22.9%)	3 (6.3%)
Fatigue	12 (25.0%)	0
Proteinuria	17 (35.4%)	2 (4.2%)
Aspartate/Alanine aminotransferase increase (baseline)	34 (70.8%)	3 (6.3%)
Aspartate/Alanine aminotransferase increase (after lenvatinib)	41 (85.4%)	10 (20.8%)
Diarrhea	3 (6.3%)	1 (2.0%)
Decreased appetite	11 (22.9%)	0
Skin rash	10 (20.8%)	0
Abdominal pain	9 (18.8%)	0
Nausea	7 (14.6%)	0
Palmar-Plantar erythrodysesthesia	3 (6.3%)	0
Bleeding	4 (8.3%)	0

## Data Availability

Data are contained within the article.

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
