# Peer review of "The Prognostic Value of Neutrophil-to-Lymphocyte Ratio and Platelet-to-Lymphocyte Ratio in Patients with Hepatocellular Carcinoma Receiving Atezolizumab Plus Bevacizumab"

_cancers, 2022, doi:10.3390/cancers14020343_

Round 1
Reviewer 1 Report
This is an important of a very few studies in real life clinical practice for Atezo+Beva in unresectable HCC. Study limitations are stated by the authors in the end of discussion ,but ,overall the data are important for clinicians in the field.
A minor point for the authors is to cite in introduction or discussion the recent review by Arvanitakis, K.; Mitroulis, I.; Germanidis, G. Tumor-Associated Neutrophils in Hepatocellular Carcinoma Pathogenesis, Prognosis, and Therapy. Cancers 2021, 13, 2899. https://doi.org/10.3390/cancers13 122899. In chapter 5 it reviews all current information for the Significance of Neutrophil-to-Lymphocyte Ratio in Hepatocellular Carcinoma as a Prognostic Marker (Also table 2).
Also some type errors have to be corrected i.e line 287 '' PLT < 230 than in those with PLR ≥ 230''
Reviewer 2 Report
General comments: As an overall analysis, this manuscript displayed an interesting study performed on 48 unresectable HCC patients receiving anti-PD-L1 therapy combined with anti-VEGF monoclonal antibody assessing the NLR and PLR as a prognostic value in progressing-free survival (PFS) rate. The manuscript demonstrated strengths and weak points according described below.
Strength:
The authors confirmed the efficacy and safety triggered by a combined approach between atezolizumab and bevacizumab in unresectable hepatocellular carcinoma patients, leading to a significant enhancement in the objective response rate and disease control rate. Also, they provide evidence between NLR/PLR and clinical outcomes, suggesting these two parameters as a prognostic value for predicting PFS in this specific cohort. In addition, the most important evidence is correlated with the impact promoted on PFS induced by combined therapy between atezolizumab and bevacizumab as first-line treatment.
Weakness:
Although the manuscript has been developed on uHCC patients' real-world clinical practice, some weak aspects could be considered before it will be accepted for publication, such as originally/novelty, and methods adequality described, and lastly, results description. These comments are described below.
The first and most important weak aspect is correlated with the originally/novelty, once this work did not bring a significant innovation, because previous studies have been similar aims, such as (Shen A., et al., 2021. Initial Experience of Atezolizumab Plus Bevacizumab for Unresectable Hepatocellular Carcinoma in Real-World Clinical Practice); (Dharmapuri S, et al., 2020. Predictive value of neutrophil to lymphocyte ratio and platelet to lymphocyte ratio in advanced hepatocellular carcinoma patients treated with anti–PD‐1 therapy); (Personeni N, et al., 2017. Prognostic value of the neutrophil-to-lymphocyte ratio in the ARQ 197-215 second-line study for advanced hepatocellular carcinoma); (Finn, RS, et al., 2020. Atezolizumab plus Bevacizumab in Unresectable Hepatocellular Carcinoma); (Galle PR, et al., 2021. Patient-reported outcomes with atezolizumab plus bevacizumab versus sorafenib in patients with unresectable hepatocellular carcinoma (IMbrave150): an open-label, randomized, phase 3 trial); (Zheng J, et al., 2017. Neutrophil to Lymphocyte Ratio and Platelet to Lymphocyte Ratio as Prognostic Predictors for Hepatocellular Carcinoma Patients with Various Treatments: a Meta-Analysis and Systematic Review). Thus, although this work studied a specific cohort (uHCC patients treated with atezolizumab plus bevacizumab), some aims are similar to those published previously.
The second point is correlated with methods adequately description, since the authors missed methods for description ROC analysis for NLR and PLR. The description of this method is crucial to the understanding of assessment and validation of the cut-off value of NLR and PLR as tool prognosis predicting.
Lastly, as the authors described in materials and methods that all patient responses were measured using dynamic computed tomography or magnetic resonance imaging of the liver, maybe would be crucial for these images to be included as supplemental material.
Round 2
Reviewer 2 Report
The authors answered satisfactorily all comments, highlighting the NLR and NLP as a prognostic value for predicting PFS on atezolizumab and bevacizumab in unrespectable hepatocellular carcinoma patients. As for originality/novelty, the manuscript will be contributing as the first study to assess this specific cohort. However, although the authors had inserted the images from dynamic computed tomography or magnetic resonance imaging as supplemental materials, still they didn't this reference on results descriptions. So, would be interesting if the authors could insert a description for these results in the text
